# Retrospective Analysis of Official Data on Anthrax in Europe with a Special Reference to Ukraine

**DOI:** 10.3390/microorganisms11051294

**Published:** 2023-05-16

**Authors:** Tamara Kozytska, Marwa Bassiouny, Olha Chechet, Diana Ordynska, Domenico Galante, Heinrich Neubauer, Gamal Wareth

**Affiliations:** 1Friedrich-Loeffler-Institut, Institute of Bacterial Infections and Zoonoses, Naumburger Str. 96a, 07743 Jena, Germany; 2State Research Institute for Laboratory Diagnostics and Veterinary and Sanitary Expertise, State Service of Ukraine for Food Safety and Consumer Protection, 30, Donetska St., 03151 Kyiv, Ukraine; 3Istituto Zooprofilattico Sperimentale della Puglia e della Basilicata, Anthrax Reference Institute of Italy, Via Manfredonia 20, 71121 Foggia, Italy; 4Institute of Infectious Diseases and Infection Control, Jena University Hospital, Am Klinikum 1, 07747 Jena, Germany

**Keywords:** anthrax, outbreaks, cases, WOAH, FAO, Europe, Ukraine

## Abstract

Anthrax is an acute infectious zoonotic disease caused by *Bacillus anthracis* that mostly affects grazing livestock and wildlife. Furthermore, *B. anthracis* is considered one of the most important biological agents of bioterrorism that could also be potentially misused in biological weapons. The distribution of anthrax in domestic animals and wildlife in Europe with a particular focus on Ukraine as a country of war was analyzed. Between 2005 and 2022, 267 anthrax cases were registered at the World Organization of Animal Health (WOAH) in animals in Europe, including 251 cases in domestic animals and 16 in wildlife. The highest numbers of cases were recorded in 2005 and 2016 followed by 2008, and the highest numbers of registered cases were reported from Albania, Russia, and Italy. In Ukraine, anthrax is currently a sporadic infection. Since 2007, 28 notifications were registered, with isolates mainly from soil samples. The highest number of confirmed anthrax cases was registered in 2018, and Odesa, which is close to Moldova, had the highest number of cases, followed by the Cherkasy region. The presence of thousands of biothermal pits and burial grounds of fallen cattle nationwide favors the re-emergence of new foci. Most confirmed cases were in cattle; however, single cases were confirmed in dogs, horses, and pigs. Further investigation of the disease in wildlife and in environmental samples is needed. The genetic analysis of isolates, investigation of susceptibility to antimicrobial compounds, and determination of virulence and pathogenicity factors are required in this volatile region of the world for awareness raising and preparedness.

## 1. Introduction

Anthrax is a serious, notifiable, non-contagious, toxin-mediated zoonosis. It is caused by *Bacillus anthracis* (*B. anthracis*), a globally distributed, Gram-positive spore-forming bacterium that has been classified as a category “A” biological agent that poses a massive destructive threat to human health and safety if used in bioweapons [1,2]. In the past, it has also been used in bioterroristic attacks in the USA [3,4]. *B. cereus* strains containing anthrax toxin genes causing severe systematic illnesses resembling anthrax have been found in the past. Thus, it was proposed to also use the term “anthrax” for such cases [1]. Anthrax is characterized by a ferocious clinical picture. It caused disease with considerable mortality in the sixteenth century, resulting in more than 60,000 deaths among humans and livestock all across Europe, but it has been prevalent for thousands of years with sporadic appearances [5]. The disease commonly causes high mortality and severe illness in a wide range of hosts, particularly domestic and wild herbivores, but other mammals such as carnivores or pigs and bird species are also affected [6,7]. Active foci are known from China, Kazakhstan, North America, Australia, sub-Saharan Africa, most European countries, and the Anatolian peninsula [7]. In humans, anthrax causes three main syndromes based on the route of infection [8]. Cutaneous anthrax occurs when *B. anthracis* spores enter through injured skin, resulting in characteristic cutaneous ulcerations resembling coal (anthrax in Greek). Injection anthrax has been described among heroin users in northern European countries, resulting in a clinical picture similar to cutaneous anthrax but with a deeper infection that may include myositis [9,10]. The inhalational form occurs when *B. anthracis* spores are inhaled. It is characterized by toxin-induced cell damage and cell death and finally results in systemic clinical illness and septic shock. Ingestion of food or water contaminated with spores results in a gastrointestinal form characterized by mucosal ulceration and bleeding [8,11]. The reservoir is contaminated soil. In the ruminant hosts, camels or equines, sudden death occurs within a few hours after showing difficult breathing, depression, and unconsciousness accompanied by the oozing of dark blood from all orifices [8,12], resulting in environmental contamination, as spores are formed when bacteria contact air. These spores are the origin of new cases even after decades. The most widely vaccine used for the prevention of anthrax worldwide in animals is the Sterne 34F2 vaccine, which was developed by Sterne in 1937 [13], and strain 55, (an equivalent pXO2-derivative) which is also the active ingredient of the current livestock vaccine in Central and Eastern Europe. The recommended dose by the World Organization of Animal Health (WOAH) as mentioned in the “OIE Terrestrial Manual 2018” should contain a minimum of 2–10 × 10^6^ culturable spores/dose for horses, cattle, and buffaloes, and 1–5 × 10^6^ for sheep, goats, and pigs. Protection is supposed to continue for at least one year and an annual booster dose is recommended.

Anthrax is endemic to several regions of the world, including southern and eastern Europe. Every year, 20,000 to 100,000 cases in animals occur worldwide, and around 1.8 billion people are at direct risk as they live in anthrax-prone areas, particularly in Africa, Europe, and Asia [7]. In Europe, several countries report yearly animal anthrax cases. The disease is endemic in Italy, with sporadic cases during summer every year [14]. A large outbreak in 2004 in the southern region of Italy involved several animal species, i.e., cattle, sheep, goats, horses, and deer [15]. In the last few years, several small outbreaks were also reported in cattle in Germany [16] and Sweden [17]. In general, North and Middle European countries only have a few sporadic cases, while anthrax remains heavily common in some countries of the Mediterranean basin such as Turkey, Italy, Spain, Greece, and the Balkan countries. Ukraine is a large Eastern European country surrounded by seven endemic countries. In Russia, anthrax killed thousands of reindeer in 2016 in Northwest Siberia [18]. In Romania, anthrax is sporadic in humans and associated with occupational exposure [19], while cases were reported annually in animals. Anthrax is a common zoonosis in Ukraine, and several cases occur repeatedly in certain regions. Whole genomic sequencing (WGS) of Ukrainian isolates assigned them to the Tsiankovskii subgroup of the “TransEurAsia” clade, pointing to a possible introduction of isolates to Ukraine and neighboring countries [20]. Between 1920 and 2019, a total of 24,955 anthrax cases occurred in different animal species. Currently, the epizootic situation of anthrax in Ukraine is complicated, as there are 10,000 known old burial grounds containing infected carcasses and up to 6000 supposed ones with unknown locations that are considered potential sources for new foci [21]. Therefore, the current report aims to summarize and update the knowledge on the European distribution of anthrax in domestic animals and wildlife species in Europe based on official data registered at WOAH and FAO, with a particular focus on Ukraine.

## 2. Materials and Methods

### 2.1. Official WOAH Data on Cases of Anthrax in Animals in Europe

The data about anthrax in domestic animals and wildlife species in Europe were extracted from the World Animal Health Information System (WAHIS). This is a database belonging to the World Organization of Animal Health (WOAH), through which information on various animal diseases since 2005 has been collected and registered worldwide. WAHIS collected information from the Veterinary Services of WOAH member and non-member countries through official reports submitted by the relevant Veterinary Services either as immediate notifications of disease foci or as follow-up reports every six months or annually. The following criteria were used: WAHIS > Analytics > Disease situation > World Region (Europe) > Animal Type (Terrestrial) > Disease (anthrax) > Disease status (Absent and Present) in the period 2005–2022. The numbers of registered anthrax cases in Europe are categorized according to country of origin, year of detection, and type of infected animals (domestic or wild) in Table 1. Additionally, information about anthrax in Europe and the frequency of anthrax cases for the last 10 years only was collected from the global animal disease information system “EMPRESS-i+” of the Food and Agriculture Organization of the United Nations (FAO).

### 2.2. Official Data on Cases of Anthrax in Ukraine

The reporting data of the annual reports on anthrax of the State Service of Ukraine on Food Safety and Consumer Protection (SSUFSCP) (https://dpss.gov.ua/news/v-kyivskii-oblasti-zafiksovano-vypadok-sybirky, accessed on 5 January 2023) and the State Scientific Research Institute of Laboratory Diagnostics and Veterinary and Sanitary Expertise (SSRILDVSE) were analyzed. During outbreaks, pathological material from dead animals (ear, blood from the incision of the ear, spleen, pharyngeal lymph nodes from pig corpses), samples of leather and fur raw materials, samples of animal feed, soil samples, and water samples were collected and sent directly to laboratories of SSUFSCP and SSRILDVSE for testing for *B. anthracis.* Microscopic examinations of stained blood smears with Gram stain and detection of capsules via Giemsa or Loeffler blue staining was the first line of diagnosis, followed by classical bacteriological examination. All blood samples from the ear were inoculated on nutrient media—meat–peptone broth (MPB), meat–peptone agar (MPA), and blood agar—and incubated at 37 °C for 24 h. Identification of *B. anthracis* was based on characteristic colony morphology, e.g., flat matte gray R-shaped colonies on MPA and blood agar with a loose precipitate in the form of a lump of cotton wool at the bottom, and long chains formed by Gram-positive *B. anthracis* bacilli. In doubtful cases, an examination of mobility (*B. anthracis* is non-motile), hemolytic properties (*B. anthracis* does not cause lysis of sheep erythrocytes), and the presence of a capsule were performed. Laboratory mice were inoculated with a subcutaneous dose of 0.1–0.2 mL of sample material in physiological solution on the surfaces of their backs on the day of receipt. Animals that died within 1–10 days were dissected, and cultures were made from blood samples of the heart, spleen, liver, and exudate at the site of injection of the test material. Serologic studies were also performed, which included an Ascoli precipitation reaction using anthrax serum, extract from the material, and standard anthrax antigen (for positive control). The final diagnosis of anthrax in animals was established when a culture with properties characteristic of the pathogen *B. anthracis* was isolated from pathological or biological material, and/or at least one of the laboratory animals injected with sample material died and *B. anthracis* was positively cultured from the organs of the dead animal.

The data were also taken from the official websites of Ukrainian government agencies such as Dniprovskyi District State Administration in the city of Kyiv, an official internet portal in the Ukrainian language, https://dnipr.kyivcity.gov.ua/news/13065.html, accessed on 5 January 2023; the State Institution “Public Health Center of the Ministry of Health of Ukraine”, official internet portal in Ukrainian language, https://www.phc.org.ua/news/v-ukraini-zafiksovano-vipadok-sibirki-vognische-lokalizovano-nebezpeki-rozpovsyudzhennya, accessed on 5 January 2023; and the State Institution “Public Health Center of the Ministry of Health of Ukraine”, official internet portal in Ukrainian language, https://www.phc.org.ua/news/v-odeskiy-oblasti-zareestrovano-vipadok-sibirskoi-virazki, accessed on 5 January 2023. 

## 3. Results

### 3.1. Anthrax Cases in Animals in Europe

Currently, the incidence of anthrax in Europe is low. Cases occur only occasionally and locally. A large-scale analysis of the World Organization for Animal Health data on the spread of anthrax in Europe among domestic and wild animals for the period 2005–2022 was conducted. Anthrax cases were registered between 2005 and 2022 in 25 countries of the European continent (Table 1). The total number of registered anthrax cases was 267, including 251 cases among domestic animals and 16 among wildlife species. As shown in Figure 1, the temporal distribution of anthrax cases in the last 18 years shows that the largest numbers of anthrax cases were recorded in 2005 and 2016 (*n* = 22 each), followed by 2008 (*n* = 21), then 2014 (*n* = 20), 2012 (*n* = 19), 2011 (*n* = 18), 2006 and 2017 (*n* = 17 each), and 2007 and 2015 (*n* = 16). The spatial distribution of anthrax cases in Europe (Figure 2 and Figure 3) during 2005–2022 shows that the highest numbers of registered anthrax cases were reported in Albania (*n* = 28), Russia (*n* = 24), and Italy (*n* = 23), followed by Romania (*n* = 20), France (*n* = 20), and Moldova (*n* = 18). The lowest numbers of anthrax cases during this period were registered in Belarus, Finland, and Switzerland, with only one case in domestic animals in each country. Most anthrax cases (*n* = 28) in domestic animals were reported from Albania. During the study period, 24 cases were registered in Russia; 20 each in France and Italy; 19 in Romania; 15 in Moldova, Greece, and Montenegro; 14 in Hungary; and 12 in Bulgaria, Croatia, and Ukraine. A few sporadic cases of anthrax in domestic animals were reported from Bosnia and Herzegovina (*n* = 8), North Macedonia and Serbia (*n* = 7), and Sweden and Germany (*n* = 4). The smallest numbers of anthrax cases among domestic animals were reported from Slovakia and Spain (*n* = 3), Poland, Slovenia, and the United Kingdom (two cases each), and only one case each was reported from Belarus, Finland, and Switzerland. In the past 18 years (2005–2022), cases of anthrax were also recorded in wildlife species. In total, 16 cases of anthrax were registered in wildlife species (Table 1). Of them, three cases were reported in Italy (2006, 2007, 2008), three in Moldova (2005, 2008), two in Bulgaria (2005), two in Hungary (2014), and one case each was reported in Greece (2008), Romania (2010), Serbia (2011), Sweden (2016), Ukraine (2021) and the United Kingdom (2006).

According to the global animal disease information system “EMPRESS-i+” of the Food and Agriculture Organization of the United Nations (FAO), Anthrax cases were reported in almost all European countries. The frequency of reported anthrax cases in Europe over the last ten years (2013–2023) fluctuates, and the highest numbers of reported cases were seen in 2016, 2018, and 2021 (Figure 4).

### 3.2. Anthrax Cases in Animals in Ukraine Based on Official Data

Ukraine is a large Eastern European country bordering seven other European countries, i.e., Poland, Slovakia, Hungary, Romania, Moldova, Russia, and Belarus, with the shores of the Black Sea and the Sea of Azov to the south (Figure 5). Based on the reporting data of the State Service of Ukraine on Food Safety and Consumer Protection (SSUFSCP), we analyzed anthrax cases and the presence of *B. anthracis* in domestic animals and environmental samples, respectively, from 2007 to 1 October 2022. Over the past 16 years, specialists of SSUFSCP state laboratories in Ukraine have examined 169,808 samples via microbiological tests to detect *B. anthracis*, and 28 cases were detected (Table 2). Of the 28 positive cases reported in Ukraine, 12 were reported in samples from soil, 10 from cattle, 2 from small ruminants, and 1 each from a horse, pig, the environment, and pathological materials (Table 2). The largest number of positive anthrax samples was registered in 2018 (*n* = 11). Of these, seven samples were from soils, three were from cattle, and one was from environmental objects. In 2012, eight isolates were recovered from five samples from soils, two from cattle, and one from a dog. It is worth mentioning that during the analysis period, *B. anthracis* was isolated mainly from soil (*n* = 12) and cattle (*n* = 10) samples. Only one case of anthrax was reported in pigs in 2016 from the Kharkiv region, one from a horse in 2021 in the Ternopil region, and one from a dog in 2012 in the Zaporizhzhya region (Table 3). In the territory of Ukraine, there are 24 regions and the Autonomous Republic of Crimea. The largest number of positive samples from 2007 to October 2022 was registered in the Odesa region (*n* = 12), with seven from the soil, four from cattle, and one from environmental objects. The Cherkasy region adds six isolates, of which five were from soils and one from cattle, followed by the Zaporizhzhia, Sumy, and Kharkiv regions, with two positive isolates reported from each.

The official data collected by SSUFSCP contain information from 2007 to 1 October 2022; however, the SSUFSCP announced the last case of anthrax in Ukraine in October 2022, i.e., a case of a goat from the Kyiv region, on its official website (https://dpss.gov.ua/news/v-kyivskii-oblasti-zafiksovano-vypadok-sybirky, accessed on 1 February 2023). According to information published on 24 October 2018 on the official website of the Dniprov District State Administration in the city of Kyiv (https://dnipr.kyivcity.gov.ua/news/13065.html, accessed on 1 February 2023), there were also two large outbreaks that occurred before 2005. In 1999, a large outbreak including 53 animals was reported from the Kherson region, and another from the Kyiv region affecting 70 animals was reported in 2001.

### 3.3. Anthrax Cases in Humans in Ukraine

We also analyzed the data from the official websites of regional state administrations, the website of the Public Health Center of the Ministry of Health of Ukraine, and the SSUFSCP for anthrax cases in humans in the period from 1999 to 2022 (Table 4). Between 1999 and 2000, 35 cases of anthrax were registered. The highest number of cases was seen in 1999 (*n* = 14), followed by 2001 (*n* = 9), 2018 (*n* = 5), three cases in 2004, and one case each in 2003, 2008, 2012, and 2020. Anthrax cases were most frequently registered in the Kherson region with eight cases, followed by the Kyiv region with seven cases, six in the Odesa region, four in the Vinnytsia region, three in the Chernivtsi and Cherkasy regions, and one each in Zaporizhzhia, Rivne, Kharkiv, and Mykolaiv regions. The samples were obtained from clinical cases and the diagnoses were based on isolation and/or real time-PCR. In all cases of human anthrax, people were ill with the cutaneous form of anthrax.

## 4. Discussion

### 4.1. Anthrax in European Countries

Anthrax is present in most parts of the world, but the frequency of cases varies. Its spores can remain latent in the soil for long periods and are activated when the soil surface is disturbed, for example by flooding, heavy rain, landslides, or excavation. Therefore, the disease reappears and infection occurs when spores are subsequently ingested by animals alongside grass from pastures. According to the registered information at WOAH, the highest numbers of anthrax cases in animals were reported from Albania, Italy, and Russia, followed by Romania, France, and Moldova. Anthrax is an endemic disease in Albania in the animal populations known since the last century. In 1992, around 150 cases of anthrax were reported in animals [22]. The genotyping of *B. anthracis* strains recovered from soil samples and dead animals showed very low diversity, and all belonged to the lineage A major subgroup A.Br.008/009 (or TEA strains), emphasizing the evolution of a local common ancestral strain in Albania [23]. Despite the authorities assuming that they vaccinated 85–100% of the animals in zones of risk, at least one or two cases are reported every year in animals (Table 1). The disease continues to be a significant human health problem as the information from the Public Health Authorities reports that the number of anthrax cases in humans is higher than the number of confirmed cases in animals [12]. Albania entered a trade deal with the European Union (EU) in 2006. However, to increase trade with the EU, Albanian authorities have to work to eradicate the major livestock diseases threatening Albanian farms, e.g., anthrax, brucellosis, and tuberculosis [24]. Therefore, there is currently a national control program for anthrax, but an anthrax-free status has not yet been achieved [24]. In Italy, anthrax is an endemic disease, with sporadic cases almost every year. However, a large outbreak occurred in 2004 involving several domestic animals and wildlife species, i.e., 81 cattle, 15 sheep, 9 goats, 11 horses, and 8 deer, in the southern region of Basilicata. Frequent cases have occurred in the same region, additionally indicating that viable spores remain in soils at outbreak sites for several years [15]. The whole genome sequencing (WGS) analysis of a strain from a new focus in the Abruzzo region and strains from Central-Southern Italy showed high genomic similarity with the Italian TEA (Trans-Eurasian) strains [25]. Another phylogenetic study using CanSNPs revealed that 231 Italian strains belonged to the sublineage A.Br. 008/009, also known as the TEA group [14], which is also predominant in Albania. Human cases are rare in Italy; however, a few occupational cases have been reported from personnel, e.g., owners of flocks and slaughterhouse workers [26,27,28]. On the contrary, anthrax is a serious problem for public health and veterinary services in Russia. Around 40–60,000 cases of this infection in animals and 10–20,000 cases in people with 25% mortality were reported annually nationwide at the beginning of the last century [29]. In 1979, 69 people died of inhalational anthrax in the Soviet Union after an accidental release of *B. anthracis* spores at a military facility in Sverdlovsk [30]. Currently, the epidemiological situation of anthrax is worrisome in Russia [31], as there are more than 35,000 anthrax stationary foci in Russia, where one or more anthrax cases were registered in animals and/or humans [29,32], with a large number of cases in the administrative territories of Siberia and southern Russia [33]. Thus, it is not astonishing that Russia is among the countries with a high incidence of cases in Europe and the impact of climatic change on the emergence of new foci over there to anthrax-free regions cannot be ruled out [18]. In Germany, anthrax is a rare disease and cases mainly occur in southern Bavaria after heavy rains. Cases in cows in 2009 and 2021 were the result of graing on the same pasture, evidencing the continued activity of old anthrax foci [16]. A similar situation has been reported in other European countries. In Sweden, cases of anthrax in cattle were reported in 2011 and 2013 in a nature reserve that was a historical burial site for anthrax-fallen cattle in the mid-1940s [17]. WGS indicated the clonality of strains from both outbreaks and thus that viable spores may remain in some sites for a few decades [34]. It is worth mentioning that the data extracted and the number of cases identified using the EMPRESS database do not match that of the WOAH database, as cases of anthrax in animals should be reported primarily to the WOAH, while the FAO is primarily concerned with food safety issues. Additionally, the FAO global search engine EMPRES-i+ combines both official and unofficial cases. Therefore, the FAO-OIE-WHO Tripartite Collaboration was developed to ensure consistency and harmonization.

### 4.2. Anthrax Cases in Ukraine

According to the Ministry of Health in Ukraine, there are more than 13,500 foci in Ukraine where anthrax may occur, as there are biothermal pits and cattle burial grounds with anthrax-infected animals buried. Thus, anthrax cases can occur anywhere in Ukraine (https://zn.ua/ukr/UKRAINE/v-ukrayini-ponad-13-tisyach-vognisch-sibirskoyi-virazki-290068.html, 20 February 2023). At the end of the last century, a soil focus of anthrax was spotted in Kyiv province [35]. Twelve positive soil samples were found in Odesa and Cherkasy in 2018 and 2012, respectively. Ukraine is a hyperendemic country, with repeated cases occurring in certain regions. However, deep knowledge of the geno- and phenotypes of local isolates is missing. WGS of a few Ukrainian isolates assigned them to the Tsiankovskii subgroup of the “TransEurAsia” clade. Thus, anthrax may have been introduced to Ukraine and neighboring countries occasionally at unknown times, forming a coherent endemic population [20]. There is an obvious data drift in the data collected from focal points in Ukraine and notifications from WAHIS of the WOAH. Only 12 cases were registered in the WOAH, while 15 cases were recorded at the national focal points. This highlights inconsistencies in international reporting of notifiable diseases. Blood screening studies of wild boars between 2011 and 2013 showed that anthrax infections in wild boars could be detected up to 35 km away from domestic anthrax cases and more than 400 km away from previous anthrax notifications in wild boars [36]. An independent sylvatic cycle can be assumed to represent a steady risk of infection for humans and free-ranging farm animals. A unique case was registered in a domestic dog, which died of anthrax in August 2012 in a yard in the village of Voznesenka, Melitopol district, Zaporizhzhya region. This was the first confirmed case of anthrax in a dog in Ukraine and was characterized by mild clinical manifestations of the disease but a short time span from the onset of the first signs to death [37].

The international availability of information on anthrax cases from Ukraine is scarce, as most information is announced in the Ukrainian language only. Very few articles have been published on anthrax in Ukraine, and thus data acquisition had to be performed by screening almost all data announced on the official governmental websites. For example, in August 2012, a single case in cattle was recorded in the village of Voznesenka, Melitopol district, Zaporizhzhya region: laboratory tests showed that a one-and-a-half-year-old heifer was infected with anthrax. The owners of the animal sold its meat in the retail network. The meat of this animal came into contact with 36 people. The remains of the infected cow were also fed to dogs, and after a few days, one of them died. The Zaporizhzhia regional laboratory of veterinary medicine published this epidemiological investigation online (https://tsn.ua/ukrayina/sibirka-na-zaporizhzhi-lokalizovana.html, accessed on 20 February 2023). In October 2018, five people who participated in the slaughter of cattle in the village of Meniailivka, Odesa region, were hospitalized with specific symptoms of skin disease, and one of the five hospitalized patients was confirmed to have anthrax by the Ministry of Health of Ukraine. The patient suffered from the skin form of the disease, which reacts to antibiotics well. *B. anthracis* was found in the meat of the slaughtered sick animal and in the soil in the yard where the animal was slaughtered (https://odesa.consumer.gov.ua/uk/294-informatsiya-pro-vipadok-zakhvoryuvannya-na-sibirku-u-s-mikolajivka-novorosijska-saratskogo-rajonu-odeskoji-oblasti, accessed on 20 February 2023). In August 2020, the reference laboratory of the Public Health Center detected *B. anthracis* DNA via PCR in a sample of a man engaged in the purchase of animals for further sale in the Odesa region. The patient had fever and a blister on the skin turned into an ulcer. He did not seek medical help immediately. Only after four days following his return did the patient consult a family doctor (https://kamyanomostivska-gromada.gov.ua/news/1597746511/, accessed on 20 February 2023). In September 2021, a pony died in the zoo of Topilche Park within the municipal enterprise “Association of Parks of Culture and Recreation of Ternopil” (Ternopil). During the autopsy of the animal, signs of pneumonia, hemorrhagic gastroenteritis, and lymphadenitis were found, and the enlargement and rounding of the edges of the spleen raised the suspicion of anthrax. According to the results of laboratory examination of the pathological samples, anthrax was diagnosed (https://dpss-te.gov.ua/golovni-novini/u-ternopoli-zareestrovano-vipadok-sibirki-v-zookutku-parku-topilche-zaginuv-infikovanii-poni-spetsialisti-derzhprodspozhivsluzhbi-rovodiat-epizootichni-zahodi, accessed on 20 February 2023).

Ukraine is surrounded by seven endemic countries, among them Russia, Romania, and Moldova, which are listed among the countries with a high number of registered cases in Europe (Table 1). In Romania, anthrax is sporadic in humans and is mainly associated with occupational exposure [19,38,39]. In the same context, contact with sick or fallen cattles is the major source of infection in humans in Ukraine, and contact with contaminated soil is the second most frequent cause of infection [21]. The highest number of Ukraine cases was reported in Odesa, which is close to Moldova. This disease has been documented as endemic in Moldova since the last century [40,41,42], and at least one case was registered every year. In Moldova, Belarus, and the Transcaucasian republics (Armenia, Georgia, Tajikistan), the epizootic and epidemic situation regarding anthrax is characterized by limited cases among animals and sporadic cases of human disease. The anthrax situation continues to be unfavorable in Kyrgyzstan, Turkmenistan, Uzbekistan, and Tajikistan [43]. Despite the geographic closeness of these countries, the phylogenetic analysis of *B. anthracis* strains isolated between 1935 and 2019 from Russia and other border countries such as Ukraine, Azerbaijan, Georgia, Armenia, and Moldova showed significant genetic diversity [44]. This is because the Russian strains are more linked to Asian strains as a result of the conquests of the Mongols that extended from China to Eastern Europe or as a result of the wool and leather trade in this region through the Great Silk Road that was laid through the lands of the North Caucasus [44]. In Ukraine, anthrax is currently a sporadic infection, and its epidemic situation is not obvious.

As a precautionary measure against anthrax, the mandatory vaccination of farm animals has been introduced, which has significantly reduced incidence over time. Mass vaccination of susceptible livestock against anthrax started in Ukraine in the early 1920s of the last century, when 21,246 animals were infected between 1923 and 1924. In 1924–1925, 2,448,600 animals were vaccinated, and between 1927 and 1928, the total number of vaccinated animals was 16,823. The State Department of Veterinary Medicine of the Ministry of Agriculture of Ukraine adopted the instructions on Measures for the Prevention and Control of Animal Anthrax at the beginning of this century. On 25 January 2000, the mandatory preventive immunization of all susceptible livestock and protection of animals from infection was implemented [45]. Live spore vaccines have been used for the past 20 years, for example, a live spore vaccine made from the SB strain manufactured by the State Enterprise “Sumy Biological Factory”, Ukraine, and from the Sterne 34F2 strain manufactured by the Kherson State Enterprise—Biological Factory, Ukraine, Kherson [45]. Compulsory vaccination is carried out at any time of the year on all animal farms regardless of the presence of infectious diseases on the farms and is administered only subcutaneously. Sheep and goats are vaccinated in the middle third of the neck or inner thigh with a dose of 0.5 mL while horses, cattle, deer, camels, donkeys, and fur-bearing animals are vaccinated only in the middle third of the neck with a dose of 1 mL and pigs are vaccinated in the area of the inner thigh or behind the ear with a dose of 1 mL. All young animals are vaccinated starting at 3 months of age and foals at 6 months. A booster dose, a second dose of re-exposure to immunization after the primary dose, is administrated to young animals after 6 months, while adults are immunized only once a year [45]. Over the past 40 years, the highest numbers of cases were observed in 1979 (33 cases), 1989 (*n* = 32), and 1994 (*n* = 33), after which there was a steady decline in the number of cases. An increasing number of detected anthrax cases in a certain time span might be due to the presence of nonimmunized animals as a result of gaps in vaccination programs or due to an increase in the number of samples tested [21]. In 2008, 2009, 2011, and 2013, there were no reported cases of anthrax in animals in Ukraine. Against the background of a decrease in the number of cases, as well as a reduction in the number of livestock susceptible to anthrax, it is also necessary to consider the favorable natural conditions for the spread of this dangerous disease near cattle cemeteries, which number up to 13,500 in Ukraine and are potential sources of *B. anthracis* spores. Recently, there is news circulating that several Russian soldiers may have contracted anthrax spores while digging trenches in the Zaporozhye region (https://promedmail.org/promed-post/?id=8709701), accessed on 8 May 2023. There are a large number of unburned anthrax-infected cattle graves in Ukraine. Any digging in these places will spread infection not only among the occupiers but also among the surrounding populations.

In conclusion, hygienic disposal of fallen animals and large-scale sampling of soil at sites of disposal and decontamination will help to reduce the risk of spore spread to animals. We also think that effective real implementation of the disease information system at the national and internal levels is crucial for controlling environmental contamination and ensuring that contaminated meat does not enter the market. Moreover, genomic analysis of a large number of isolates is essential to understand the phylogeography of *B. anthracis*, investigate its susceptibility to antimicrobial compounds, and analyze its virulence and pathogenicity factors.

## Figures and Tables

**Figure 1 microorganisms-11-01294-f001:**
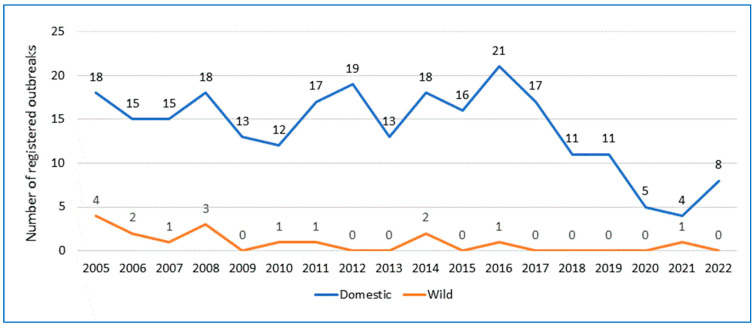
Chronological distribution of anthrax cases in Europe in domestic animals and wildlife species between 2005–2022based on official data of the WOAH.

**Figure 2 microorganisms-11-01294-f002:**
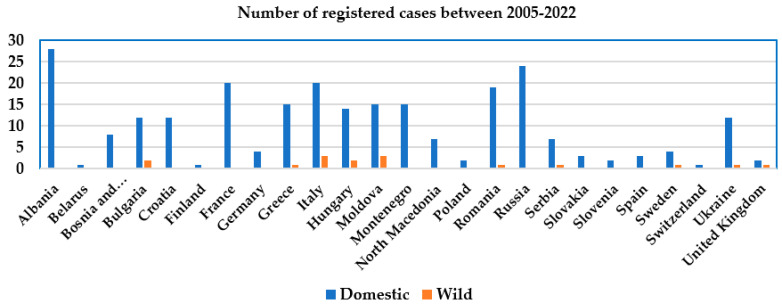
Geographical distribution of anthrax cases in Europe in domestic animals and wildlife species between 2005–2022 based on official data of WOAH.

**Figure 3 microorganisms-11-01294-f003:**
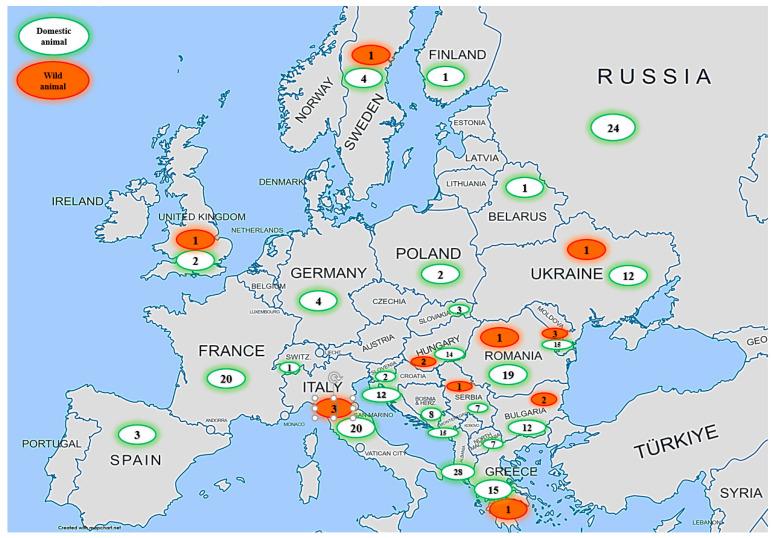
“Map of anthrax cases in European countries from 2005–2022”. The map was generated using the open-source software MapChart (https://www.mapchart.net/europe.html, accessed on 5 January 2023). Numbers in the map refer to number of cases in each country.

**Figure 4 microorganisms-11-01294-f004:**
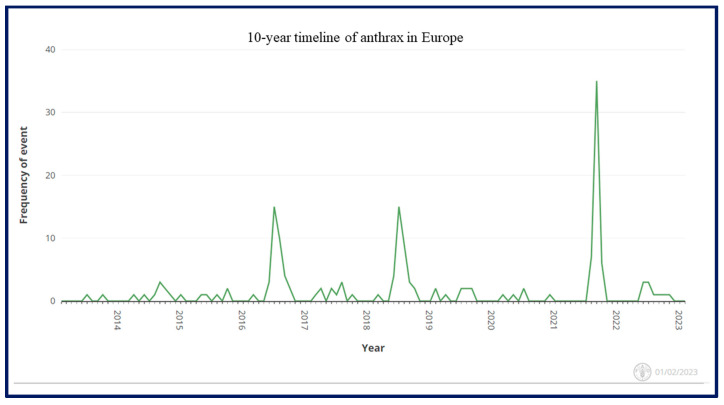
The timeline frequency of anthrax in Europe over the last ten years (2013–2023) according to the EMPRESS-i of FAO (accessed on 1 February 2023).

**Figure 5 microorganisms-11-01294-f005:**
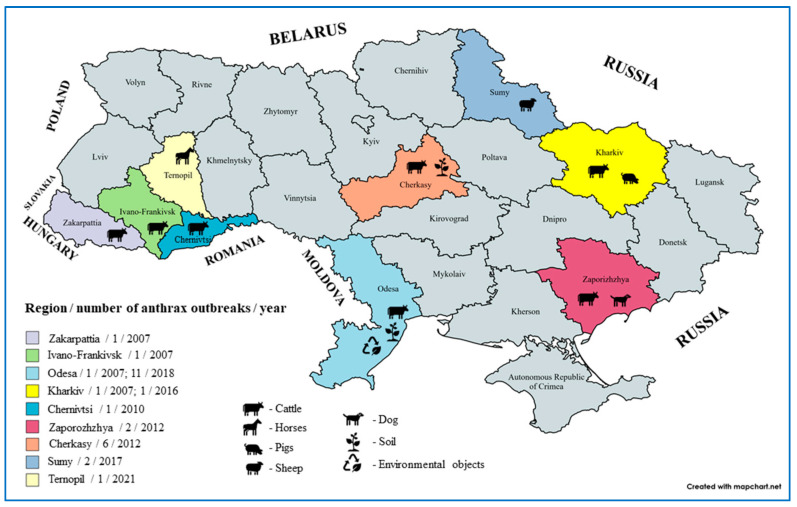
Map of Ukraine showing numbers, locations, and sources of anthrax cases in the last 16 years (2007–2022). The map was generated using the open-source software MapChart (https://www.mapchart.net/europe.html, accessed on 1 February 2023).

**Table 1 microorganisms-11-01294-t001:** Reported anthrax cases in domestic animals and wildlife in Europe between 2005 and 2022 based on WOAH data. Numbers ≥ 20 are colored with red background.

Country	2005		2006		2007		2008		2009		2010		2011		2012		2013		2014		2015		2016		2017		2018		2019		2020		2021		2022		Total	
	Domestic	Wild	Domestic	Wild	Domestic	Wild	Domestic	Wild	Domestic	Wild	Domestic	Wild	Domestic	Wild	Domestic	Wild	Domestic	Wild	Domestic	Wild	Domestic	Wild	Domestic	Wild	Domestic	Wild	Domestic	Wild	Domestic	Wild	Domestic	Wild	Domestic	Wild	Domestic	Wild	Domestic	Wild
Albania	1		2		2		2		2		2		2		2		2		2		2		2		1		2		2								28	0
Belarus																													1								1	0
Bosnia and Herzegovina	1						2				1		1				1		1						1												8	0
Bulgaria	2	2	1								1				2		1		1		2		2														12	2
Croatia	1		1		1								2		2				2												2		1				12	0
Finland							1																														1	0
France	1		1		2		2		2								1						2		2		2		2				1		2		20	0
Germany									1						1				1																1		4	0
Greece	2		1				2	1	2		1		2		2								1		2												15	1
Italy	2		2	1	1	1	1	1					1		1				1		2		2		1		2		2		2						20	3
Hungary	1				1				1		1				1		1		2	2	2		1				2		1								14	2
Moldova	2	2	1		1		1	1	1		1		2		1		1		1		2		1														15	3
Montenegro											2		2		1		2		2		2		2		2												15	0
North Macedonia					1																1		2		2				1								7	0
Poland																			2																		2	0
Romania	2		2		1		2		2			1	1						1				2		2		1		1		1				1		19	1
Russia	2		2		2		2		2		2		1		2		2		1		1		2				1		1						1		24	0
Serbia					1		1						2	1	1		1								1												7	1
Slovakia											1				1				1																		3	0
Slovenia							1														1																2	0
Spain																																	1		2		3	0
Sweden							1						1				1						1	1													4	1
Switzerland																									1												1	0
Ukraine	1		1		2										2								1		2		1						1	1	1		12	1
UK			1	1																	1																2	1
Total	18	4	15	2	15	1	18	3	13	0	12	1	17	1	19	0	13	0	18	2	16	0	21	1	17	0	11	0	11	0	5	0	4	1	8	0	251	16
	22		17		16		21		13		13		18		19		13		20		16		22		17		11		11		5		5		8		267	

**Table 2 microorganisms-11-01294-t002:** Report of anthrax cases in animals in Ukraine between 2007 and 1 October 2022 based on the data of the State Service of Ukraine on Food Safety and Consumer Protection. Data show the number of samples tested and confirmed positive cases.

Region	Source of Samples																	
	Horses		Cattle		Small Ruminants		Pigs		Soil		Pathological Materials ^3^		Leather		Environmental Objects		Total for Region	
	Total ^1^	Pos. ^2^	Total	Pos.	Total	Pos.	Total	Pos.	Total	Pos.	Total	Pos.	Total	Pos.	Total	Pos.	Total	Pos.
Autonomous Republic of Crimea (2007–2013)	17	0	1209	0	84	0	2439	0	560	0	13,316	0	6236	0	4	0	23,865	0
Vinnytsia	52	0	623	0	74	0	650	0	2858	0	181	0	0	0	6	0	4444	0
Lutsk	4	0	127	0	25	0	119	0	1585	0	15	0	0	0	11	0	1886	0
Dnipro	48	0	345	0	71	0	336	0	3805	0	36	0	1	0	4	0	4646	0
Donetsk	24	0	4484	0	515	0	2203	0	919	0	520	0	0	0	208	0	8873	0
Zhytomyr	55	0	288	0	56	0	188	0	2171	0	999	0	0	0	1375	0	5132	0
Zakarpattia	4	0	121	1	4	0	94	0	962	0	436	0	0	0	0	0	1621	1
Zaporizhzhya	41	0	1162	1	144	0	1817	0	1900	0	105	1	760	0	151	0	6080	2
Ivano-Frankivsk	3	0	220	1	9	0	62	0	1807	0	9	0	0	0	0	0	2110	1
Kyiv	29	0	586	0	6	0	303	0	1713	0	301	0	119	0	36	0	3093	0
Kirovograd	16	0	263	0	142	0	865	0	2267	0	0	0	0	0	0	0	3553	0
Luhansk	194	0	1870	0	620	0	6028	0	4747	0	7	0	0	0	3	0	13,469	0
Lviv	43	0	588	0	99	0	802	0	5663	0	755	0	3935	0	4505	0	16,390	0
Mykolaiv	14	0	177	0	23	0	182	0	1561	0	1	0	102	0	0	0	2060	0
Odesa	60	0	872	4	606	0	1907	0	3267	7	14,040	0	1007	0	292	1	22,051	12
Poltava	28	0	702	0	92	0	1170	0	7617	0	3803	0	0	0	56	0	13,468	0
Rivne	10	0	306	0	28	0	196	0	6348	0	3	0	0	0	0	0	6891	0
Sumy	26	0	236	0	49	2	241	0	3154	0	533	0	40	0	89	0	4368	2
Ternopil	6	1	111	0	3	0	34	0	2814	0	3	0	0	0	8	0	2979	1
Kharkiv	14	0	768	1	730	0	1982	1	1850	0	46	0	0	0	3	0	5393	2
Kherson	4	0	90	0	37	0	48	0	2592	0	59	0	0	0	66	0	2896	0
Khmelnytsky	58	0	763	0	56	0	263	0	2444	0	58	0	10	0	0	0	3652	0
Cherkasy	12	0	125	1	42	0	127	0	4199	5	249	0	0	0	293	0	5047	6
Chernivtsi	14	0	137	1	71	0	96	0	2046	0	264	0	0	0	12	0	2640	1
Chernihiv	22	0	601	0	35	0	122	0	2383	0	33	0	0	0	5	0	3201	0
TOTAL	798	1	16,774	10	3621	2	22,274	1	71,232	12	35,772	1	12,210	0	7127	1	169,808	28

^1^: total number of tested samples, ^2^: number of confirmed cases, ^3^: pathological materials include organs and tissues from dead animals.

**Table 3 microorganisms-11-01294-t003:** Spatiotemporal distribution of anthrax cases in animals in Ukraine in the last 16 years (2007–2022).

Report 2007–2022			
Number of Positive Cases (*n* = 28)	Year of Report	Region	Source
1	2007	Zakarpatskyi region	Cattle
1	2007	Ivano-Frankivsk region	Cattle
1	2007	Odesa region	Cattle
1	2007	Kharkiv region	Cattle
1	2010	Chernivtsi region	Cattle
1	2012	Zaporizhzhya region	Cattle
1	2012	Zaporizhzhya region	Dog
1	2012	Cherkasy region	Cattle
5	2012	Cherkasy region	Soil
1	2016	Kharkiv region	Pigs
2	2017	Sumy region	Small ruminants
3	2018	Odesa region	Cattle
7	2018	Odesa region	Soil
1	2018	Odesa region	Environmental samples
1	2021	Ternopil region	Horse

**Table 4 microorganisms-11-01294-t004:** Spatiotemporal distribution of anthrax cases in humans in Ukraine between 1999 and 2020 based on official websites of regional state administrations.

Year	Region	Host	Number	References
1999	Kherson	Human	8	Dniprovskyi District State Administration in the city of Kyiv, information published on 24 October 2018, on the official internet portal in the Ukrainian language https://dnipr.kyivcity.gov.ua/news/13065.html, accessed on 1 February 2023
1999	Vinnytsia	Human	4	
1999	Cherkasy	Human	2	
2001	Kyiv region	Human	7	
2001	Zaporizhzhya	Human	1	
2001	Rivne	Human	1	
2003	Kharkiv	Human	1	State Institution “Public Health Center of the Ministry of Health of Ukraine”, information published on 2 October 2018 on the official internet portal in Ukrainian language https://www.phc.org.ua/news/v-ukraini-zafiksovano-vipadok-sibirki-vognische-lokalizovano-nebezpeki-rozpovsyudzhennya, accessed on 1 February 2023
2004	Chernivtsi	Human	3	
2008	Mykolaiv	Human	1	
2012	Cherkasy	Human	1 *	
2018	Odesa	Human	5 *	
2020	Odesa	Human	1 *	State Institution “Public Health Center of the Ministry of Health of Ukraine”, information published on 12 August 2020 on the official internet portal in Ukrainian language https://www.phc.org.ua/news/v-odeskiy-oblasti-zareestrovano-vipadok-sibirskoi-virazki, accessed on 1 February 2023

*: diagnosis was based on RT-PCR.

## Data Availability

Not applicable.

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
