# Peer review of "Retrospective Analysis of Official Data on Anthrax in Europe with a Special Reference to Ukraine"

_microorganisms, 2023, doi:10.3390/microorganisms11051294_

Round 1

Reviewer 1 Report

There are not many recent publications describing anthrax surveillance and a current paper on this topic would be of interest. However, I recommend the authors consider suggestions to revise this current draft to improve the content and flow.

Consider restructuring the manuscript so that it flows more smoothly, and the data presented from different sources are comparable. There are some hotspot data available (i.e., Bagamian et al, DOI: 10.1089/vbz.2013.1521).  It would be useful to map the focal points even to the rayon level, if possible, so that this information is put into context with the overall geography of the country.

Consider framing the state surveillance data in the same context as the WAHIS and Empressi, then get into more on the specifics – i.e., first describe outbreak counts, then specify number of positive tests. This will make it easier for the reader to compare to the rest of the region. Do the data reported to WAHIS match those reported within the country - this is included in the discussion, thank you for mentioning this discrepancy.

Present the same time frame from EMPRESSi as for WAHIS, or present 10 years ending in the same year as the WAHIS data (present data through 2022, if possible)

The first section of the results should be moved to methods. Also, the year range in the results does not match the year range given in the methods.

Be clear what data are being presented from WAHIS – in line 7 of the results, “The total number of registered anthrax outbreaks was 267, including 251 cases among domestic animals and 16 among wildlife species.” – be clear on whether these are outbreaks or cases. Or define what you mean when you use each term. Additionally the EMPRESSi data presented uses the term "event". Either change all terms to the same wording, or define each if they are different. 

Suggest removing most of the counts of the outbreaks by country in the first paragraph of the results, since it is also in the table. It is easy to get lost in the numbers, so summarizing and referring to the table is more effective.

The large number of events in Figure 4 are not reflected in the WAHIS data. This discrepancy should be discussed, including if there are known reporting time lags.

Additionally, countries that do not report outbreaks in wildlife- do they have wildlife surveillance programs to even identify cases, or are there truly no cases in wildlife? This should be discussed. 

Table 2: there is no legend for the colors. Either remove the colors or add a legend.

Figure 2: recommend making a map to show distribution, so can visually see how countries near each other compare, and then remove Figure 3.

Table 3: be clear whether these counts are individual cases or outbreaks potentially involving more than 1 animal. I assume from Table 2 that these counts are individual cases.

Table 4: Recommend adjusting the methods to include these reports prior to 2007 when describing what data were used. It is confusing to read the methods and then see these additional data sources brought in. Also, the reader may question where other data could be if they are familiar with the anthrax literature, such as the >400 cattle cases discussed in Bagamian et al.

Section 3.3, line 56: this data source should be moved to methods section

The extensive discussion could be shortened for brevity, and the cases described may be more appropriate in the results section. Consider including a comment on the potential effects of disruption of routine vaccination. 

Reviewer 2 Report

For this review, the authors extracted information from the World Animal Health Information System, and from the EMPRESS global animal disease information system to catalog the number of anthrax cases in wildlife and domestic animals in Europe, with a specific emphasis on the situation in Ukraine. For the latter, State Service of Ukraine on Food Safety and Consumer Protection and the State Scientific Research Institute of Laboratory Diagnostics and Veterinary and Sanitary Expertise reports were analyzed. Examples of human cases were also incorporated. This information should prove useful to interested in the global distribution of Bacillus anthracis and levels of disease. However, there was a disparity between the incidence of animal anthrax cases identified by the EMPRESS database and that of the WOAH database. A better discussion of how to interpret these results, and the likely reasons behind the differences should be added to the review.

Specific comments:

1.            It is unclear from this review why the EMPRESS numbers for European cases per year (Figure 4) differ so markedly from the WOAH numbers (Figure 1). For example, in 2021 the latter indicated 5 cases while the former indicted 35. Which set is the more reliable picture of anthrax cases in Europe and why?

2.            Section 3.3 and Table 4:  It is likely that the human cases were mostly cutaneous in nature. Is that correct or was that type of information unavailable.

3.            Section 4.1, lines 38-40:  Need a reference or references for this claim. The claim seems inconsistent with the numbers reported later in this section.

Author Response

Response to reviewer #2 comments:

For this review, the authors extracted information from the World Animal Health Information System, and from the EMPRESS global animal disease information system to catalog the number of anthrax cases in wildlife and domestic animals in Europe, with a specific emphasis on the situation in Ukraine. For the latter, the State Service of Ukraine on Food Safety and Consumer Protection and the State Scientific Research Institute of Laboratory Diagnostics and Veterinary and Sanitary Expertise reports were analyzed. Examples of human cases were also incorporated. This information should prove useful to those interested in the global distribution of Bacillus anthracis and levels of disease. However, there was a disparity between the incidence of animal anthrax cases identified by the EMPRESS database and that of the WOAH database. A better discussion of how to interpret these results and the likely reasons behind the differences should be added to the review.

Specific comments:

  1. It is unclear from this review why the EMPRESS numbers for European cases per year (Figure 4) differ so markedly from the WOAH numbers (Figure 1). For example, in 2021 the latter indicated 5 cases while the former indicted 35. Which set is the more reliable picture of anthrax cases in Europe and why?

 Respond: Thank you very much for the time you have devoted to our manuscript. We agree with you that the number of cases identified by the EMPRESS database does not match the WOAH database, as the FAO global search system EMPRES-i+ combines both official and unofficial cases. Cases of anthrax in animals should be reported to the WOAH. The time line (Figure 4) is generated automatically by the search engine, using all the data together. We will use a screen copy for our publication.

  1. Section 3.3 and Table 4:  It is likely that the human cases were mostly cutaneous in nature. Is that correct or was that type of information unavailable.

 Respond: Thank you for the good addition, indeed, human exposures were cutaneous, p. 11 line 71-72.

  1. Section 4.1, lines 38-40:  Need a reference or references for this claim. The claim seems inconsistent with the numbers reported later in this section.

Respond: A link has been added on page 14 to address the comment.

Reviewer 3 Report

This is a comprehensive analysis of official data on anthrax in Europe with a special reference to Ukraine. I don't have major comments for improvement. However, moderate english changes are required. Some typos, such as "60,000 death", "20,000 to 100,000 cases", and table 4, need to be corrected.

Author Response

Response to reviewer #3 comments:

This is a comprehensive analysis of official data on anthrax in Europe with a special reference to Ukraine. I don't have major comments for improvement. However, moderate English changes are required. Some typos, such as "60,000 death", "20,000 to 100,000 cases", and table 4, need to be corrected.

Respond: Thank you for pointing this out. The "60,000 death", and "20,000 to 100,000 cases" has been corrected on p.2 line 7, 42, and in Table 4.